# Fracture Resistance of Glass-Fiber-Reinforced Direct Restorations on Endodontically Treated Molar Teeth with Furcal Perforation

**DOI:** 10.3390/polym17030370

**Published:** 2025-01-29

**Authors:** Ecehan Hazar, Ahmet Hazar

**Affiliations:** 1Department of Endodontics, Faculty of Dentistry, Zonguldak Bülent Ecevit University, 67600 Zonguldak, Turkey; 2Department of Restorative Dentistry, Faculty of Dentistry, Zonguldak Bülent Ecevit University, 67600 Zonguldak, Turkey; dt.ahmethazar@yahoo.com.tr

**Keywords:** Bioblock, fracture resistance, furcal perforation, horizontal glass fiber, pericervical dentin, short-fiber-reinforced composite, unidirectional glass fiber

## Abstract

This in vitro study evaluated short-fiber-reinforced composite materials and fiber-reinforced restorations of endodontically treated molars with furcal perforation. The endodontic treatment and mesio-occlusal–distal cavity preparation of 126 two-rooted mandibular third molars were performed. Eighteen non-perforated teeth were restored with resin composite as the control group. Furcal perforations and repair were performed on 108 teeth that were divided into six experimental groups: resin composite (RC), everX Flow (EXF), everX Posterior (EXP), Bioblock (BB), modified transfixed (MT), and horizontal glass-fiber (HGF) groups (n = 18). Fracture resistance tests were performed at an angle of 30◦ using a universal testing machine under static loading, and fracture patterns were classified. Welch’s analysis of variance, Pearson chi-square, and Tamhane post hoc tests (*p* = 0.05) were used to analyze the data (*p* = 0.05). The highest fracture resistance values were seen with the HGF (596.305 N), followed by MT (540.365 N), BB (477.906 N), EXP (476.647 N), EXF (414.462 N), control (413.811 N), and RC (335.325 N) groups (*p* < 0.001). There was no significant difference between the BB and EXP groups or between the EXF and control groups (*p* > 0.05). In terms of the dominant fracture pattern, the HGF and MT groups were repairable and possibly repairable, whereas the control, RC, and EXP groups were unrepairable. The EXF and BB groups were almost equally divided between possibly repairable and unrepairable. Restorations using horizontal fiber techniques and short-fiber-reinforced materials increased the fracture resistance of endodontically treated teeth with furcal perforation.

## 1. Introduction

The prognosis of teeth during endodontic treatment is affected by the loss of hard tissues, such as the pulp chamber roof, marginal ridges, and pericervical dentin (PCD). Endodontic access preparation reduces the relative stiffness by 5% and increases functional cusp deflection, which jeopardizes the tooth’s resistance to fracture [1]. If the cavity dimensions increase due to caries or trauma and the remaining wall thickness decreases, cusp deflection increases. It has been reported that mesio-occlusal–distal (MOD) cavity preparation reduces the structural strength of the tooth by approximately 63% [2]. With the recent increasing interest in minimally invasive endodontics, studies are being conducted to explore the effects of access cavity design, root canal preparation protocols, and PCD on endodontically treated teeth (ETT). The PCD, which forms the neck of the tooth, is in the region 4 mm above or 4 mm below the crestal bone [3]. Protecting this area is crucial for maintaining the durability of the biomechanical function of the tooth by providing resistance against tooth fractures [4].

Perforations occur for a variety of reasons, including extensive caries, resorption, or iatrogenic factors [5]. The PCD structure, which plays an important role in transmitting functional stresses from the crown to the apex, can be damaged by furcal perforations during endodontic treatments. The biomechanical response of the tooth might differ from that of an intact tooth even if the perforation is repaired due to the loss within this area. finite element energy analysis (FEA) studies performed on 3D models reported that the accumulation and distribution of stress are influenced by furcal perforation, which leads to an increased risk of fracture even if these areas are repaired with calcium-silicate-based cement [6,7].

After endodontic treatment, it is essential to restore the tooth in a timely manner to protect the remaining tooth tissue and prevent microleakage [5]. Endodontically treated posterior teeth can be restored with direct restorations using a post and core, resin composite, and amalgam materials, or indirect restorations, such as partial or full crowns using ceramic and hybrid materials. Resin composites are preferred for direct restorations due to their adhesive and esthetic properties and appropriate elastic modulus [8]. Flowable resin composites are preferred due to their low viscosity, which provides easier manipulation and placement. However, the decreased filler content of this material results in reduced polishability and physical properties of restoration. However, these materials have considerably lower filler contents compared with condensable resin composites, which results in lower polishability and an irregular surface morphology [9]. To overcome these disadvantages of flowable resin composites without compromising their positive properties, a new class of flowable materials called injectable resin composites has emerged. The mechanical properties of these materials are superior to predecessor flowable resin composites. Furthermore, due to their uniform spreading ability, their marginal adaptation is better than that of condensable resin composites [10].

Fiber reinforcement of resin composites increases the toughness and fracture resistance of restorations [11]. Short-fiber-reinforced composites (SFRCs) composed of millimeter-sized, randomly oriented E-glass fibers and a specific semi-interpenetrating polymer network, which provides favorable adhesion properties to the material, have been developed as a specialized material for the restoration of high-stress-bearing areas in molars [12]. everX Posterior (EXP) is composed of 140 μm sized E-glass microfibers and 700 nm sized barium glass fillers [13]. The short-fiber structure of EXP provides enhanced strength and reinforced mechanical properties to restorations by preventing the initial formation and propagation of cracks [13]. In 2019, everX Flow (EXF), a short-fiber-reinforced material, was introduced. EXF consists of shorter, thinner fibers that help prevent the difficult handling disadvantages of its predecessor, EXP [14].

Post–core applications enhance the retention of the restoration in ETT without coronal dentin. Fiber posts have favorable physical properties due to their modulus of elasticity and are commonly used materials. During application, the post space needs to be prepared before the polymerized, prefabricated fiber posts are cemented into the root canal. However, the removal of radicular dentin and loss of PCD structure during the preparation of this post space further weaken the remaining tooth and increase the risk of tooth fracture or root perforation [15]. Various conservative techniques have been sought to preserve the PCD of ETT in contemporary dentistry. The placement of fiber posts from the buccal wall to the lingual wall perpendicular to the long axis of the tooth is one such technique. A recent systematic review reported that horizontal glass-fiber post (HGF) application increased the fracture resistance of composite restorations [16]. A clinical study stated that this technique is more economical in comparison to a full crown preparation and preserves the natural tooth structure [17].

The modified transfixed (MT) technique is another option that has been previously reported [18]. With the MT technique, the fracture resistance of endodontically treated maxillary premolars increases. This is performed by horizontally bonding everStick C&B continuous unidirectional long fibers (GC Corporation, Tokyo, Japan) between the buccal and lingual walls perpendicular to the long axis of the tooth. everStick C&B have pre-impregnated E-glass fibers containing linear polymer phases, which form a semi-interpenetrating polymer network after polymerization. This structure reinforces the teeth by providing anisotropic mechanical properties [19].

In addition to complications that might occur during the preparation of the post space, the inadequate adaptation of prefabricated posts to the cervical region of the root canal results in failure due to excessive luting cement and biomechanically inappropriate positioning of fibers within the root canal [20]. However, reports suggest that these disadvantages can be eliminated by preparing a custom post suitable for the root canal for which the Bioblock (BB) technique has been introduced [21]. In this technique, the root canal (intraradicular post) and the coronal space (core and dentin replacement) are filled with SFRC at horizontal increments of 4–5 mm thickness.

The restoration of ETT is crucial for fracture-related failure, necessitating reinforcement by specialized techniques and materials. During treatment planning of such complex cases, it is important to predict the biomechanical response of teeth with structural losses, such as perforations. To the best of our knowledge, the effect of restoration techniques on fracture resistance in teeth with furcal perforation repaired with calcium silicate cement has not been tested. This in vitro study aimed to evaluate the reinforcement effects of the modified transfixed technique, horizontal glass-fiber technique, and Bioblock technique, and the use of SFRC materials for ETT with an MOD cavity and furcal perforation. The null hypotheses were that (1) furcal perforation would decrease the fracture resistance of teeth, (2) the fracture resistance of fiber-reinforced restorations would increase, and (3) there would be no differences in fracture patterns of restorations when using different glass-fiber techniques.

## 2. Materials and Methods

Table 1 presents the manufacturers and compositions of the materials used in this study.

### 2.1. Tooth Selection

Ethical approval of this study was obtained from the Non-Interventional Clinic Research Ethics Committee of Zonguldak Bülent Ecevit University (protocol number: 2024/12). In this study, two-rooted mandibular third molar teeth were obtained from patients aged 18–30 years. The sample size was calculated using G*Power (version 3.1.9.7, Kiel University, Kiel, Germany) and one-way analysis of variance (ANOVA) in the findings of a previous study [18]. The effect size (f) was 0.43, the type I error (α) was 0.05, and the statistical power (1-β) was 0.95. The required total sample size for this study was 126, and the required sample size for each group was 18. The teeth were stored in distilled water at 37 °C until use. To minimize possible confounders, teeth with an unusual morphology on the occlusal surface and roots were excluded. The teeth were examined under a dental operating microscope (EZ4W, Leica Microsystems, Milton Keynes, UK) to exclude those with cracks or fractures after the removal of debris and soft tissue remnants. Radiographs were taken of each tooth, and those with a dentin–cementum thickness of 2–2.5 mm in the furcation area and a pulp chamber height of 3–4 mm were included. Teeth with calcification or resorption lesions were excluded from this study.

The teeth were assessed according to their coronal and root dimensions. The dimensions were measured with a digital caliper (SC-6 digital caliper, Mitutoyo Corporation, Tokyo, Japan), and the measurements were statistically analyzed (one-way ANOVA, α = 0.05) to establish whether the teeth were similar to each other. Standardized endodontic cavities were prepared using an electric motor (NLX Nano electric motor, NSK, Tokyo, Japan) and cylindrical diamond burs (G837/018, Dia Tessin, Vanetti, Gordevio, Switzerland) under water cooling by a single operator (EH). Root canals were determined using size 10 K-files (Maillefer SA, Ecublens, Switzerland), and teeth with apically completed development, two independent canals in the mesial root, and a single canal in the distal root were included in this study. In total, 126 teeth met the inclusion criteria and were selected (n = 18).

### 2.2. Endodontic Treatment

Size 10 K-files were inserted into the root canals until the file tip was visible and the working length for each canal was determined by subtracting 1 mm from the obtained length. The root canals were shaped with the ProTaper Gold System (Dentsply, Tulsa Dental Specialties, Tulsa, OK, USA) to a size of F2 for the mesial canals and F3 for the distal canal. Following the use of each file, 2 mL of 3% sodium hypochlorite was used to irrigate the canals. The root canals were dried using absorbent paper points after final irrigations with 17% ethylenediaminetetraacetic acid (EDTA, Sigma-Aldrich, St. Louis, MO, USA) for 1 min followed by rinsing with distilled water. The appropriate gutta-percha (PTG gutta-percha tip, Maillefer SA, Ecublens, Switzerland) and resin paste (AH Plus, Maillefer SA, Ecublens, Switzerland) was then used with the single cone technique for canal obturation (Figure 1a). The teeth were stored in distilled water at 37 °C for 7 days after the endodontic cavities were cleansed of residual root canal sealer with ethanol and sealed with a temporary filling material.

### 2.3. MOD Cavity Preparations

All teeth underwent a standardized MOD cavity preparation with an approximate depth of 5 mm (measured from the corresponding cusp tip by touching the cavity wall), a width of 3 mm (buccolingual), and a 2.5 mm wall thickness (at the base of the cavity on both vestibular and oral aspects) using an electric motor at 20,000 rpm and cylindrical diamond burs (G837/018, Dia Tessin, Vanetti, Gordevio, Switzerland) under water cooling by a single operator (AH) (Figure 1b,c). The thickness of the opposing walls, depth, and width at the cavity were continuously checked during the preparation with a 15 UNC periodontal probe (Hu-Friedy Mfg. Co., Chicago, IL, USA). The cavity walls were prepared parallel to the axis of the tooth.

### 2.4. Furcal Perforation and Repair

The temporary filling materials were removed. Except for one group, which served as a control (n = 18), cavities were prepared in the middle of the pulp chamber floor of the experimental groups of teeth using a high-speed handpiece with diamond burs under water cooling to simulate furcal perforations. Perforation cavities were formed by a single operator (EH) under water cooling using round diamond burs (SG801L/014, Dia Tessin, Vanetti, Gordevio, Switzerland). The size of all perforations was standardized to a 1.4 mm diameter using round diamond burs, and the depth of the perforations ranged from 2 to 3 mm according to the dentin–cementum thickness in the furcation area (Figure 1d).

All perforation cavities were restored incrementally using a hand plugger (Queen Instruments, Hungary) with the perforation repair material (Biodentine, Septodont, Saint-Maur-des-Fossés, France) mixed according to the manufacturer’s instructions by a single operator (EH; Figure 1e). The teeth were then stored at 37 °C in distilled water for 1 week. The root canal orifices and perforation repair material were sealed using a resin-modified glass ionomer cement (Nova Glass GL, Imicryl Dental Materials Inc., Konya, Türkiye) following the application of adhesive (Scotchbond Universal Plus Adhesive, 3M Deutschland GmbH, Neuss, Germany), except for the teeth in the BB group (Figure 1f).

### 2.5. Periodontal Ligament Simulation

As described in a previous study, a layer of elastomeric impression material (Oranwash L, Zhermack, Badia Polesine, Italy) was applied to the root surface of each tooth to simulate the periodontal ligament before embedding [22]. The teeth were then embedded in methacrylate resin (Technovit 4004, Heraeus-Kulzer, Hanau, Germany) to a depth of 2 mm below the cementoenamel junction (CEJ), to simulate the bone level.

### 2.6. Adhesive Application

Adhesive (Scotchbond Universal Plus Adhesive, 3M Deutschland GmbH, Neuss, Germany) was applied to the dentin and the etched enamel walls according to the manufacturer’s instructions, followed by polymerization of the adhesive for 20 s using an LED curing unit (Elipar S10, 3M, St Paul, MN, USA) with an output intensity of 1200 mW/cm^2^. A single operator (AH) performed the adhesive procedure. After the adhesive application, the teeth were divided into six groups according to the direct restorative techniques to be performed (Table 2).

### 2.7. Control Group

Teeth without furcal perforation were used in the control group (n = 18) (Figure 2a). An AutoMatrix band (Dentsply, DeTrey GmbH, Konstanz, Germany) was placed around the tooth, after which the cavities were restored with an injectable flowable composite material (G-aenial Universal Injectable, GC Corporation, Tokyo, Japan), which was applied incrementally with a 2 mm thickness of layers. Each layer was light-cured using an LED curing unit for 40 s. The interproximal walls were not built up before the restoration but were restored simultaneously with the entire cavity (Figure 2c).

### 2.8. Resin Composite (RC) Group

Eighteen teeth in which furcal perforation and repair were performed were used for the RC group (Figure 2b). An AutoMatrix band (Dentsply, DeTrey GmbH, Konstanz, Germany) was placed around the tooth, after which the cavities were restored with an injectable flowable composite material (G-aenial Universal Injectable, GC Corporation, Tokyo, Japan). The material was applied incrementally with layers of 2 mm thickness. Each layer was light-cured using an LED curing unit for 40 s. The interproximal walls were not built up before the restoration but were restored simultaneously with the entire cavity like in the control group (Figure 2d).

### 2.9. everX Flow (EXF) and everX Posterior (EXP) Groups

An injectable resin composite was used to convert the MOD cavities to Class I and build up 1.5 mm thickness proximal walls (G-aenial Universal Injectable) after placing an Automatrix band around the tooth (Figure 3a,d). The prepared cavities were restored with everX Flow (Figure 3b) or everX posterior (Figure 3e) up to 2 mm below the occlusal surface according to the experimental groups (n = 18). The remaining part of each cavity was filled with an injectable resin composite (G-aenial Universal Injectable; Figure 3c,f).

### 2.10. Modified Transfixed (MT) Group

An everStick C&B glass-fiber strip with a 1.5 mm width and 3 mm length was cut to fit the buccolingual space of the prepared cavity. A thin composite layer (G-aenial Universal Injectable) was applied to the center of the buccal and lingual inner walls, where the ends of the fiber strip were to be bonded. No further preparation was undertaken to the cavity walls. The adhesive was applied to the surface of the fiber strip, and the excess adhesive was removed using air–water spray. The fiber strip was placed horizontally on the specified area and light-cured, followed by restoration procedures. The interproximal walls were not built up before the restoration but were restored simultaneously with the entire cavity, which was performed identically to those of the RC and control groups (Figure 3g–i).

### 2.11. Horizontal Glass-Fiber (HGF) Group

Before adhesive application, an extra preparation step was undertaken to fit fiber posts for this group. Two transverse holes were prepared with round diamond burs through the buccal (SG801L/014, Dia Tessin, Vanetti, Gordevio, Switzerland) and lingual (SG801L/012, Dia Tessin, Vanetti, Gordevio, Switzerland) walls. Two glass-fiber posts (X-Post Size 2, Dentsply DeTrey, Konstanz, Germany) were fitted through the holes, protruding from the buccal and lingual walls. After the adhesive procedure, an injectable resin composite (G-aenial Universal Injectable) was used to cement the two posts in place (Figure 4a,b). The interproximal walls were not built up before the restoration but were restored simultaneously with the entire cavity, like in the control, and resin composite groups. After restorative procedures were performed, the protruding portions of the posts on the buccal and lingual surfaces were cut off (Figure 4c–e).

### 2.12. Bioblock (BB) Group

For teeth in the BB group, the root canal filling was removed piece by piece from each canal by heating it with a plugger (Fast-Pack PRO, Eighteeth, Changzhou, China) up to 3 mm below the CEJ (Figure 4f). The depth of the root canal space where gutta-percha was removed was checked with a periodontal probe. Residual root canal sealer was removed from the walls with ethanol and ultrasonic activation (VDW.ULTRA System, VDW, Munich, Germany). No additional preparation was performed with the drill to create the post space, thus preserving the anatomy of the root canal. The same adhesive treatment was applied to the cavity and prepared post space of the samples as in the other groups, except that the root canal applicator tips (Dentsply DeTrey, Konstanz, Germany) were used to apply adhesive in the post spaces. The excess adhesive resin at the bottom of the post space was eliminated using a paper point before polymerization. An injectable resin composite was used to build up proximal walls (G-aenial Universal Injectable) after placing an Automatrix band around the tooth. The cavities including the 3 mm depth post space were then restored with flowable SFRC (EXF) up to 2 mm below the occlusal surface using the horizontal layering technique (Figure 4g,h). The remaining parts of the cavities were filled identically to those of the SFRC groups with an injectable composite material (G-aenial Universal Injectable; Figure 4i). Radiographs were taken of each tooth (Figure 4j). The finishing and polishing procedures were performed using alumina discs (OptiDisc, Kerr, Bioggio, Switzerland) for all groups.

### 2.13. Fracture Resistance Test

The prepared teeth were kept in distilled water at 37 °C until they were subjected to the fracture resistance test. The teeth were then subjected to a 30° oblique compressive load in a universal testing machine (PWS-E100, Shimadzu Co., Kyoto, Japan). The load was applied to the junction of the restoration and enamel with a crosshead speed of 0.5 mm/min until fracturing occurred (Figure 5a). The failure load, defined as the load at which the tooth–restoration complex exhibited the first fracture, was recorded in Newtons (N). After the fracture test, each tooth was examined for fracture patterns under a dental operating microscope (EZ4W, Leica Microsystems, Milton Keynes, UK) at ×16 magnification and with two-examiner agreement. The fracture patterns were classified as repairable (fractures not extending below the CEJ; Figure 5b), possibly repairable (fractures extending below the CEJ, but not below the acrylic line; Figure 5c), and unrepairable (fractures extending below the acrylic line; Figure 5d). The fracture patterns of each tooth were recorded, and the percentages of the fracture patterns in the groups were calculated.

### 2.14. Statistical Analyses

Statistical analysis was conducted with IBM SPSS V23 software (SPSS Inc., Chicago, IL, USA). The Shapiro–Wilk test was used to analyze the conformity of the values to a normal distribution. Welch’s analysis of variance (ANOVA) test was used to compare the fracture resistance values between the groups. The Tamhane post hoc test was used for pairwise comparisons. The repairability of the groups after fracturing was analyzed using Pearson’s chi-squared test. The level of significance was set at *p* < 0.05.

## 3. Results

Figure 6 presents the mean and standard deviation values of the groups tested in this study. Welch’s ANOVA results suggest that the restoration techniques and materials affect the fracture resistance of restorations (*p* < 0.001). According to Tamhane post hoc test results, the highest fracture resistance values were in the HGF group (596 ± 44 N), followed by the MT (540 ± 48 N), BB (477 ± 49 N), EXP (476 ± 36 N), EXF (414 ± 25 N), control (413 ± 62 N), and the RC (335 ± 73 N) groups (*p* < 0.05). The mean fracture resistance values of the BB and EXP groups were not significantly different (*p* > 0.05). Also, there was no significant difference between the mean fracture resistance values of the EXF and control groups (*p* > 0.05).

The fracture pattern distribution for each group is presented in Figure 7. Repairability was significantly affected by the restoration technique used (*p* < 0.001). Among the groups, the highest percentages of teeth that were repairable after fracturing were in the HGF (50%), control (11.1%), and MT (11%) groups. There were no repairable fracture patterns in the RC, EXF, EXP, and BB groups (0%). The highest percentage of unrepairable teeth was observed in the RC (88.9%) and EXP (88.9%) groups. The highest percentage of possibly repairable teeth was observed in the BB and MT groups (50%).

## 4. Discussion

Endodontic treatment weakens the tooth structure and makes it more susceptible to fracturing. One of the main aims of restoring teeth after root canal treatment is to regain lost strength. Direct resin restorations allow the tooth to be treated without preparation and without sacrificing the remaining structure, and they are also more economical and less time-consuming than indirect techniques [23]. Additionally, in cases where teeth need to be kept under observation until the healing of periapical or furcal lesions can be assessed, direct restorations might be more appropriate than indirect restorations, hence the rise in the demand and popularity of affordable restorations. Therefore, the fracture resistance and patterns of direct composite resin restorations using different fiber techniques in weakened teeth treated with endodontic treatment were investigated in this study.

Root perforation constitutes an undesirable complication in endodontic treatment, compromising the structural integrity of the root and leading to the further destruction of adjacent periodontal tissues [24]. An FEA study reported that higher stress values occurred in 3D models with central furcal perforation compared with perforations in the mesial and distal regions [7]. Therefore, in this study, a perforation area was created between the mesial and distal roots to mimic weakened teeth with a questionable prognosis under occlusal forces. Introducing new and advanced materials and techniques in endodontic practice has enabled dentists to attempt to repair perforations with a conservative approach. Biodentine is a commonly used perforation repair material due to its biocompatibility, ion release characteristic, short setting time, and high push-out bond strength. Additionally, Biodentine has a compressive strength of 304 MPa, which is close to human dentine [25]. Therefore, Biodentine was used as the perforation repair material in this study given its beneficial properties.

Although there is much research on choosing appropriate biomaterials for perforation repair, there are insufficient data on restoration techniques after perforation repair [24,26]. To our knowledge, there are no studies that investigate the fracture resistance of molar teeth after coronal restoration of perforations, which has been reported in 3D models to change the stress distribution in the tooth [7]. In our study on mandibular molars, teeth with perforation repair (RC group) showed lower fracture resistance than teeth restored without perforation (control). Therefore, we accepted our first hypothesis. This finding supports FEA studies in the literature and demonstrates the necessity of strengthening the coronal restorations of ETT that have damaged PCD regions [6,7]. According to the results of this study, all fiber-reinforced restorative techniques showed increased fracture resistance compared with the RC group. As such, we accepted our second hypothesis. In addition, when teeth with furcal perforation were restored with fiber reinforcement, they showed fracture resistances similar to (EXF) or higher (EXP, MT, BB, and HGF) than composite restorations without perforation (control). As a result, instead of applying composite materials alone for direct restorations, fiber reinforcement or the use of SFRC as a dentin replacement material could be a better solution for these perforated teeth.

In this study, the fracture resistance of restorations made with SFRCs as the dentin replacement material was significantly higher than that in the RC group. This finding is in line with two systematic reviews, one on posterior teeth with large MOD restorations and the other on ETT [27,28]. The fracture toughness of composite materials is lower than that of dentin [29]. Fracture toughness is used to describe the damage tolerance of material and can be considered a measure of fatigue resistance; thus, it is a predictor of the material’s structural performance [30]. Given the increased material volume in direct restorations, the problem of a lack of toughness becomes more evident, as with ETT with MOD cavities. The high fracture toughness of SFRCs and 3D, isotropic, randomly oriented E-glass fibers in all directions provides a better fracture resistance than in composite resins [27,29]. Randomly orientated E-glass fibers are crack-stopping; therefore, a small crack propagating through the material cannot grow further when it encounters these fibers [13]. In addition to E-glass fibers’ structural behavior, a semi-interpenetrating network resin matrix reduces the stress concentration [31]. When EXF and EXP materials were used as dentin replacement materials, EXP showed a significantly higher fracture resistance in this study. However, there are other studies reporting that EXP and EXF exhibit similar fracture resistance levels in teeth with or without root canal treatment [32,33,34]. In a recent clinical study, EXP and EXF showed similar efficacies in terms of restoration and tooth fracture in endodontically treated Class I restorations [32]. In another study, the fracture strength of molar teeth restorations made with EXP and EXF (bulkfill) used at a 3 mm thickness was reportedly similar [31]. In a study of root-canal-treated premolars with mesio-occlusal cavities, no difference in survival was found between EXP and EXP under simulated normal or parafunctional forces [32]. We suggest two possible reasons for the contradictory results. The first suggestion involves the cavity depth to which we apply the SFRC materials. In a recent study, it was reported that EXF and EXP showed a similar fracture toughness to natural dentin and that both are suitable dentin replacement materials for biomimetic restorations [35]. However, the authors reported that the depth of application of the material should be considered. They reported that EXF had greater toughness than EXP in samples prepared with 2 mm thickness, while both materials had similar toughness levels in samples prepared with 4 mm thickness. Therefore, the authors suggest that EXF is suitable for shallow cavities in terms of fracture resistance, while both can be applied in deep posterior cavities. In our study, both materials were placed in the pulp chamber (3–4 mm) and coronal section (3 mm), with a total thickness of 6–7 mm in posterior teeth. The increasing depth might have caused the materials to exhibit different physical properties by changing their fracture toughness. Similar to our research, in a study conducted on a plastic dental cavity of 5 mm depth and width, the fracture resistance of EXP was higher than that of EXF [36]. Additionally, the authors reported that EXF showed more shrinkage-induced cuspal deformation than EXP [36]. The second possible reason for the contradictory results is that shrinkage-induced cuspal deformation becomes more important in teeth that are further weakened by perforation.

In this study, three different techniques were compared in addition to dentin replacement with SFRCs. The first of these techniques is the BB technique applied to the intraradicular region to provide retention. With the development of fiber-reinforced composites, individual posts created by curing these materials in the root canals have emerged. In the BB technique, disadvantages, such as interface failure caused by adhesive cement and biomechanically inappropriate positioning of prefabricated posts, are eliminated by firmly bonding the SFRC in the root canal. Due to these features, it has been suggested that there might be a reduction in tensile stress under masticatory forces after the restoration is completed [37]. This technique allows for restoration by providing intraradicular retention without additional preparation in curved roots or irregular root canal sections, which are not considered ideal for prefabricated posts. Use of the BB technique can be facilitated clinically with the introduction of flowable bulk-fill SFRCs. Previous studies reported that the fracture resistance of restorations using this technique increased compared with those using prefabricated posts [21,38,39]. In this study, the BB technique increased the fracture resistance, which decreased with furcation perforation. According to the results of this study, the BB technique prepared using EverX flow showed a similar (EXP-packable SFRC) or superior (EXF-flowable SFRC) fracture resistance to SFRCs used as dentin replacement material. These findings are partially consistent with the results of a previous study conducted on premolar teeth [34]. The mentioned study revealed that the fatigue strength of single-rooted premolars was comparable when flowable and packable SFRC was applied with the BB technique (3 mm or 6 mm depth) or as the dentin replacement material. Although the materials used in both studies were the same, the fact that the BB group showed a better fracture resistance than the EXF group in our study might have been due to the application of the technique to three canals in double-rooted teeth. To the best of our knowledge, this is the first study to use the BB technique on multi-rooted teeth. With this technique, stress distribution might have been achieved by increasing the number of fibers in the PCD around the perforation area. The stress might have been distributed away from this weakened area and toward the outer surface of the root [40].

The second technique evaluated in this study was coronal reinforcement with horizontal glass-fiber posts. A recent systematic review reported an increased fracture resistance in premolars and molars restored with HGF [16]. The results of our study are consistent with the systematic review. Among the experimental groups, the highest fracture resistance was observed in the HGF group. A study conducted on third molars restored with two HGFs, as with our study, reported that it had higher fracture resistance than direct composite restorations [2]. The authors stated that this technique showed a similar fracture resistance to onlay restorations and could offer an alternative to indirect restorations. Contrary to our results, one study reported that the placement of HGF did not increase the fracture resistance in mandibular first molars restored with composite resin [41]. In the study, a single horizontal post was used and the compressive load was applied parallel to the long axis. This difference between these and our results might be due to the number of posts applied before the restoration, the direction of the load, or the crown size of the evaluated teeth.

Another technique evaluated in our study was coronal reinforcement with unidirectional, long glass fibers, which we introduced in our previous study [18]. In this study, the fracture resistance was higher with the MT technique using everStick C&B than in the other groups, except for HGF. Similarly, in our previous study, we reported that this technique showed a superior fracture resistance to EXF and different fiber orientation techniques for root-canal-treated maxillary premolar teeth [18]. The long, continuous, and unidirectional E-glass fibers reportedly provide great reinforcement against forces perpendicular to their long axis [19]. Cuspal deflection due to anchoring of the buccal and lingual walls can be reduced by bonding the fiber strip between the cusps. This result might be due to the material’s inherent properties of becoming strengthened when stretched and distributing occlusal forces in the same way as observed in the pulp chamber roof [18,42]. With this technique, unlike HGF techniques, thanks to its flexible structure, everStick C&B could be cut to a length appropriate to the buccolingual cavity width and bonded without groove preparation. This technique can be considered more conservative than HGF because it reduces structural loss due to groove preparation. It should be noted that fracture resistance is lower than with HGF, but this approach has potential in conservative dentistry because of its greater fracture resistance compared with other techniques. Additionally, this technique seems suitable for bonding thicker, unidirectional, long fiber strips or for bonding unidirectional, long fiber strips at different levels in ETT with high crown heights. We assume that this technique could be enhanced with the aforementioned modifications, thereby increasing its contribution to fracture resistance, but further studies are required to validate this assumption.

When fracture patterns were examined, unrepairable fracture patterns were dominant in the RC group, and no repairable fracture was observed in any of the restorations in this group. Higher rates of possibly repairable and repairable fractures were observed than in the RC group in restorations of the teeth without perforation (control group). According to our findings, although the furcal perforation was repaired with biomaterial, the fracture pattern of restorations changed for the worse. The negative effect of perforation on the fracture pattern appeared to be reduced in fiber-reinforced restorations (except for EXP). The fracture pattern in the MT and HGF groups was better than that in the control group. In the systematic review, HGF application reduced irreparable fractures in premolars, but there was no significant difference for molars [16]. The authors suggested that this situation could have been caused by the loss of an additional structure or the formation of a microcrack while preparing the post hole for two horizontal posts in molars, which might have adversely affected tooth fracture patterns. However, in our study, the fracture line was frequently seen above or 1–2 mm below the placed posts, and this group had the lowest rate of irreparable fracture patterns. A possible reason for this observation might be that the applied stress is transmitted to the holes prepared through the post instead of being transmitted to the pericervical area, resulting in fractures occurring there. A study using a single horizontal post on mandibular first molars might support our presumption, as the researchers reported that single post application did not improve the fracture pattern [41]. In our previous study on endodontically treated premolar teeth, we reported that MT and EXF showed a lower unrepairable fracture pattern compared with the non-fiber-reinforced group, in line with the current results [18]. Additionally, the rate of repairable fractures was higher in the MT group than in the EXF groups. Similarly, in this study, a higher rate of repairable fractures was observed in the MT group. Of note, although higher fracture resistance values were obtained in the EXP group compared with the EXF, RC, and control groups, the fracture pattern was predominantly irreparable, as with the RC group, and no repairable fractures were observed in any of the restorations. Better fracture patterns were observed in the EXF group than with EXP. Consistent with this finding, another study stated that the extreme resistance of EXP led to more catastrophic failures than EXF for large MOD restoration [36]. We reject our third hypothesis because fracture patterns varied in all groups depending on the material or technique used.

In this study, the effects of glass-fiber-reinforced dentin replacement, horizontal fibers, and individual post applications on the fracture resistance of furcal perforated teeth were compared using different materials. Upon evaluation, we found an interesting outcome that in teeth that had a PCD region that was weakened, horizontal reinforcement of the crown with long glass fibers provided a better fracture resistance and fracture pattern than SFRC applied to the root canal or pericervical region. This result might have occurred because of increased cusp deflection due to the removal of the pulp cavity roof for endodontic treatment and perforation in the PCD region in MOD cavities. Long glass fibers bonded horizontally above the PCD region could provide reduced deflection by connecting the cusps, acting similarly to a pulp cavity floor [42].

The fracture resistance of the restorative material when exposed to occlusal forces and its ability to strengthen the remaining tooth structures are important factors for a successful restoration. In the posterior region, forces range from 8 to 880 N during normal mastication [43]. The real mastication force (magnitude, direction, frequency, location, and duration of the force) is quite complicated and variable. Static loading mimics the parafunctional forces that occur when biting hard foods (seeds, nuts, etc.) or clenching the teeth. In a recent study, maximum bite force measurements were reported to be 333 N at the second and first molars [44]. Since the static load-fracture test is applied without fatigue testing and without the multiple directions of actual bite force, it cannot be directly related to the clinical scenario. However, after comparing the results of this clinical study [44] and our in vitro study, we propose that mandibular molars with root canal treatment and furcal perforation, especially when restored without fiber reinforcement (335 N), are at risk of fracture under the parafunctional forces that occur when biting hard foods.

One of the limitations of this study was that static load-to-fracture testing was used without fatigue testing. While static loading is crucial for evaluating fracture resistance, it is important to consider that materials might behave differently under dynamic or cyclic loads, which can lead to different failure mechanisms. Understanding the material performance in a clinical scenario requires comprehensive testing involving both static and dynamic conditions. When considering clinical conditions, preparing parallel post holes in the HGF group could be challenging for posterior teeth. Additionally, an insufficient buccal and lingual wall thickness or inadequate crown length might make using this technique difficult. With HGF or MT techniques, applying a packable composite after the fiber is bonded could be difficult in terms of manipulation. In this study, this limitation was overcome by using an injectable composite. However, the choice of core material should be a well-considered decision because the performances of materials vary [45]. Using flowable SFRC as a core material with these techniques could offer a good alternative to injectable composites. Future studies should focus on dynamic loading tests and aging procedures to better understand the behavior of materials and techniques. Additionally, finite element analysis may be performed to investigate the effect of applied techniques on the stress distribution.

## 5. Conclusions

Within the limitations of this in vitro study evaluating the effects of different fiber-reinforced materials and techniques on endodontically treated third mandibular molar teeth with furcal perforation, the following conclusions can be drawn:Loss of tissue in the pericervical dentin region increases the risk of tooth fracturing.Direct restorations using techniques such as Bioblock, modified transfixed, horizontal glass-fiber, and dentin replacement materials including everX posterior and everX Flow can compensate for the reduction in fracture resistance caused by furcal perforation.The horizontal glass-fiber technique shows a superior fracture resistance, and in cases where this technique cannot be applied clinically, the modified transfixed technique using everStick C&B could offer a conservative treatment option to increase fracture resistance.Horizontal bonding of the buccal and lingual walls with long, continuous glass fibers improves the fracture pattern more than short-fiber reinforcement techniques.

## Figures and Tables

**Figure 1 polymers-17-00370-f001:**
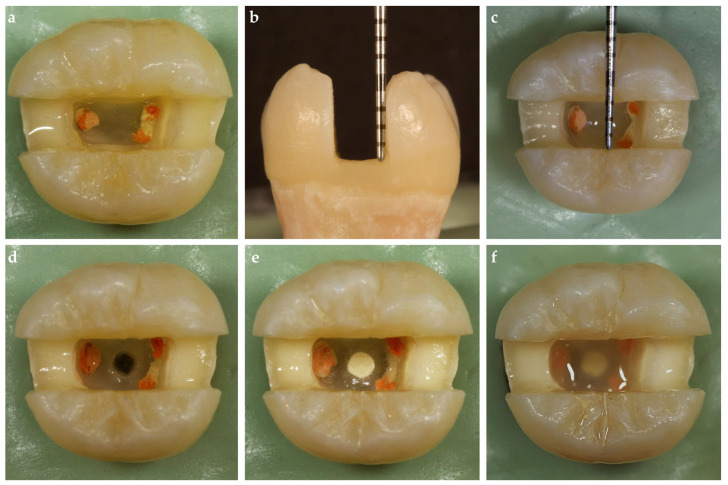
Representative preparation images. (**a**) Endodontic treatments of the teeth were performed; (**b**,**c**) measurements of the cavity dimensions were taken with a periodontal probe; (**d**,**e**) furcal perforations and repair of the perforations were performed; (**f**) root canal orifices and perforation repair material were sealed using a resin-modified glass ionomer cement.

**Figure 2 polymers-17-00370-f002:**
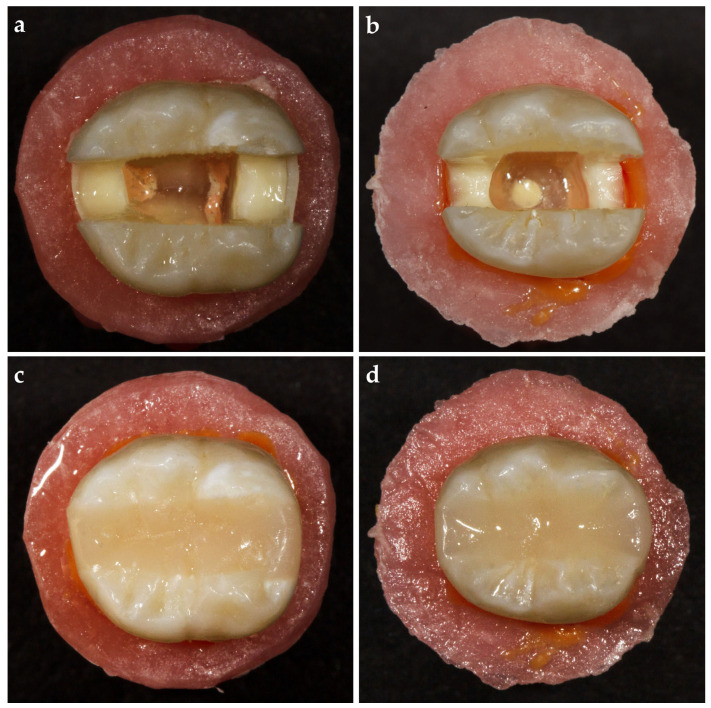
Representative preparation images. (**a**,**c**) Control group; (**b**,**d**) resin composite (RC) group.

**Figure 3 polymers-17-00370-f003:**
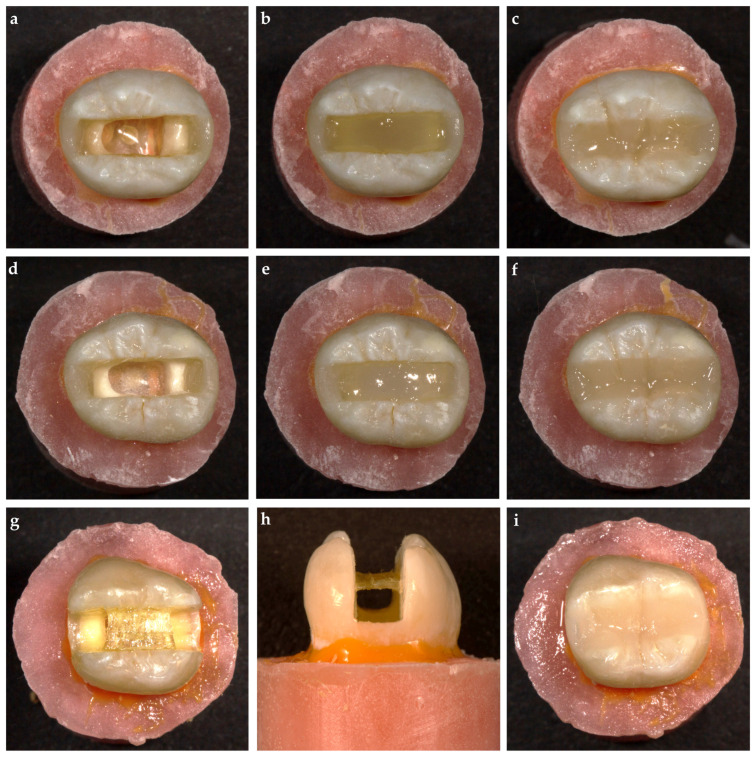
Representative preparation images. (**a**–**c**) Flowable short-fiber-reinforced composite group (EXF); (**d**–**f**) packable short-fiber-reinforced composite group (EXP); (**g**–**i**) modified transfixed technique group (MT).

**Figure 4 polymers-17-00370-f004:**
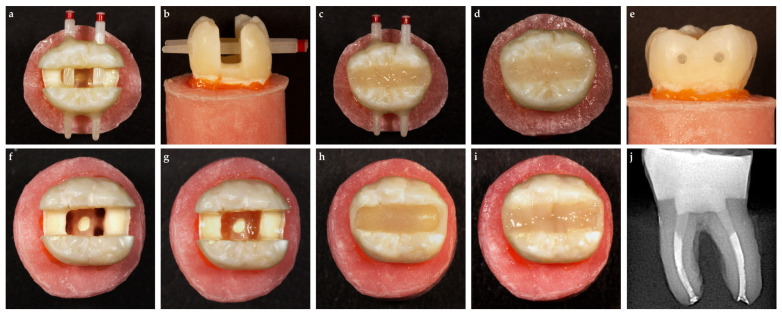
Representative preparation images. (**a**–**e**) Horizontal glass-fiber technique group (HGF); (**f**–**j**) Bioblock technique group (BB).

**Figure 5 polymers-17-00370-f005:**
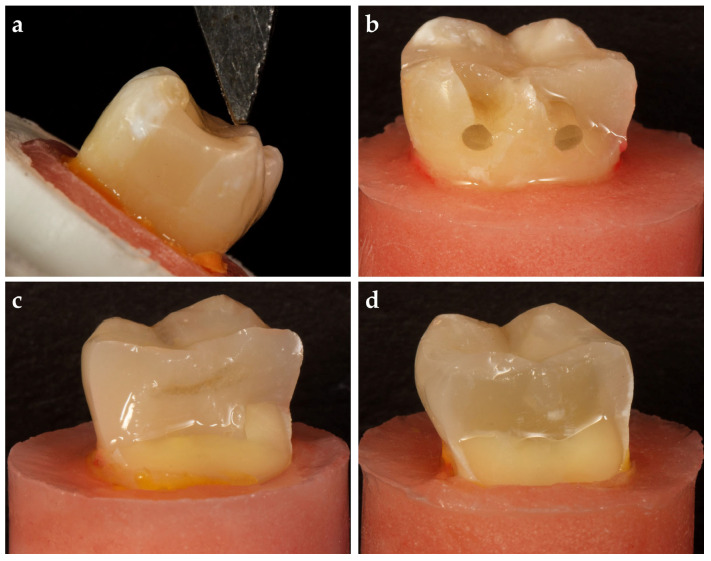
Representative image of (**a**) fracture strength test; (**b**) repairable fracture pattern; (**c**) possibly repairable fracture pattern; (**d**) unrepairable fracture pattern.

**Figure 6 polymers-17-00370-f006:**
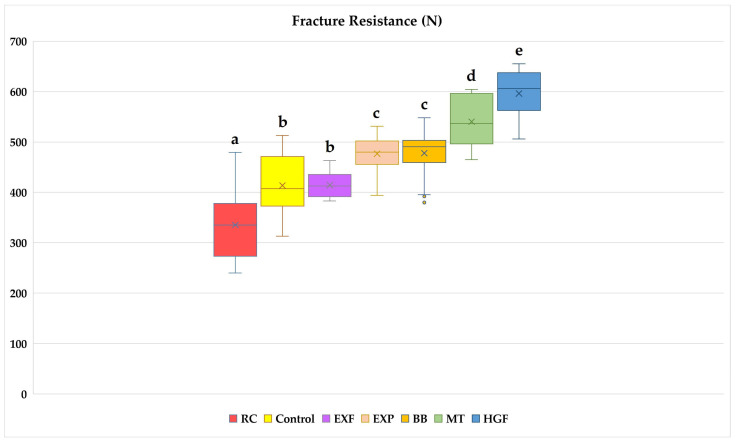
Fracture resistance values of the groups (Welch’s ANOVA test/test statistic: 53.225/*p* < 0.001). a–e: Different lowercase letters indicate significant differences between groups (Tamhane post hoc test).

**Figure 7 polymers-17-00370-f007:**
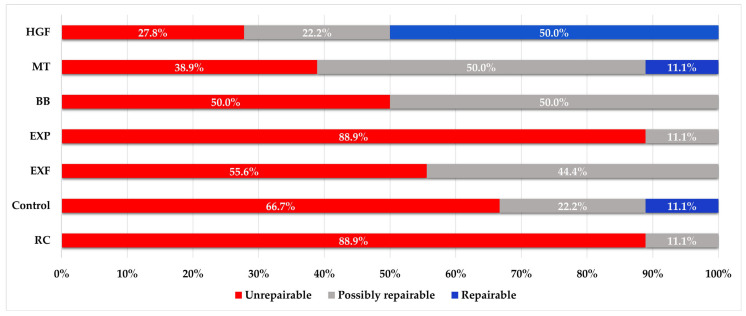
Fracture pattern distributions (%) of the groups.

**Table 1 polymers-17-00370-t001:** Materials used in this study.

Material	Manufacturer	Fillers	Matrix
G-aenial Universal Injectable, flowable composite	GC Corporation, Tokyo, Japan (Lot number: 2205201)	Silicon dioxide (SiO_2_), barium glass, 69 wt%, 50 vol%	UDMA, Bis-MEPP, TEGDMA
everX Flow™, flowable short fiber reinforced composite	GC Corporation, Tokyo, Japan (Lot number: 2106221)	Micrometer scale glass fiber filler, barium glass, 70 wt%, 46 vol%	Bis-EMA, TEGDMA, UDMA
everX Posterior™, short fiber reinforced composite	GC Corporation, Tokyo, Japan (Lot number: 2308071)	Shot E-glass fiber filler, Barium glass 74.2 wt%, 53.6 vol%	Bis-GMA, PMMA, TEGDMA
everStick^®^C&B fibers	GC Corporation, Tokyo, Japan (Lot number: 2212192)	Silanated, unidirectional glass fibers	PMMA, Bis-GMA
Scotchbond Universal Plus Adhesive	3M Deutschland GmbH, Neuss, Germany (Lot number: 10665120)	Bis-GMA, 10-MDP, 2-HEMA, Vitrebond copolymer, ethanol, water, initiators, fillers	
Biodentine™	Septodont^®^, St. Maur-des-Fossés, France (Lot number: B31840)	Cement composition: tricalcium silicate (Ca_3_SiO_5_), dicalcium silicate (Ca_2_SiO_4_), zirconium oxide (ZrO_2_), calcium carbonate (CaCO_3_) Liquid composition: hydrosoluble polymer (polycarboxylate), calcium chloride (CaCl_2_), water	

**Table 2 polymers-17-00370-t002:** Study groups tested.

Study Group	Representative Group Name	Furcal Perforation	Applied Material	Restoration Technique
Control	Control	No	G-aenial Universal Injectable	Injectable composite resin without fiber reinforcement—incremental technique
Resin composite	RC	Yes	G-aenial Universal Injectable	Injectable composite resin without fiber reinforcement—incremental technique
everX Flow	EXF	Yes	everX Flow™+ G-aenial Universal Injectable	Dentin replacement with flowable SFRC—bulk technique
everX Posterior	EXP	Yes	everX Posterior™+ G-aenial Universal Injectable	Dentin replacement with packable SFRC—bulk technique
Bioblock	BB	Yes	Everx flow+ G-aenial Universal Injectable	Radicular retention and dentin replacement with flowable SFRC—bulk technique
Modified transfixed	MT	Yes	everStick^®^C&B fibers + G-aenial Universal Injectable	Coronal, horizontal long glass-fiber application/ injectable composite resin without fiber reinforcement—incremental technique
Horizontal glass-fiber	HGF	Yes	X-Post + G-aenial Universal Injectable	Coronal, horizontal long glass-fiber application/ injectable composite resin without fiber reinforcement—incremental technique

## Data Availability

Data are contained within the article.

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
