# Peer review of "Fracture Resistance of Glass-Fiber-Reinforced Direct Restorations on Endodontically Treated Molar Teeth with Furcal Perforation"

_polymers, 2025, doi:10.3390/polym17030370_

Round 1

Reviewer 1 Report

Comments and Suggestions for Authors

The study aims to evaluate the reinforcement effects of the modified transfixed technique, horizontal glass-fiber technique, and Bioblock techniques, and the use of SFRC materials for ETT with MOD cavity and furcal perforation. The article describes an interesting topic.

Please find my comments below.

Title and abstract

The information on in vitro study should be added.

Introduction

It could be shortened.

Materials and methods

Please add a calculation of sample size and power.

Please add a figure or a table presenting study groups.

Describe in detail the technique of restoring control group (as separate paragraph) and study groups with respect to interproximal walls.

When and how the interproximal walls in MT and HGF groups were reconstructed.

Line 258 – add the type of alcohol

Was the occlusal surface characteristics the same for all groups? If yes, how it was standardized? It could have had a significant influence on the results if in some samples the enamel was covered with composite resin and in others was not.

Study group with typical reconstruction using post placed in root canal should be added.

Study groups differed with regard to technique and materials used therefore it is impossible to indicate the reason for the difference in the results.

The cavity design is far different from the one performed in clinical scenarios.

Results

Present figure 6 as box and whiskers plot.

Discussion

It should be shortened.

Line 401-402: ‘Additionally, the authors reported that EXF showed more shrinkage-induced cuspal deformation than EXP.’ – add a reference or information how it was measured

Are fracture resistance values obtained for control and study groups observed in clinical conditions? What is clinical significance of the study?

Limitations of the study: what other examination methods could be employed for the further analysis?

What are future perspectives?

Conclusions

Conclusions are presented as summary of the results. 

Comments on the Quality of English Language

English grammar should be double checked, i.e. line 60, 74, etc.

Author Response

Dear Reviewer,

Thank you for your valuable contribution. We believe that your comments and correction requests will improve both this study and our future work.

The requested revisions and corrections have been made and highlighted in yellow.

Comments 1: The information on in vitro study should be added.

Response 1: The term in-vitro was added to the abstract section.

Comments 2: Please add a calculation of sample size and power.

Response 2: Information on how sample size was calculated and power analysis was added to the materials and methods section.

Comments 3: Please add a figure or a table presenting study groups.

Response 3: Information on how the study groups were prepared was added as an additional table in the manuscript (Table 2).

Comments 4: Describe in detail the technique of restoring control group (as separate paragraph) and study groups with respect to interproximal walls.

Response 4: Control and resin composite groups were separated and written in different paragraphs as requested.

Comments 5: When and how the interproximal walls in MT and HGF groups were reconstructed.

Response 5: Fiber-reinforced composites such as everX Flow and everX Posterior are not recommended for use in regions that are open to the oral environment. Since we use these materials as dentin replacement materials, we only applied them in the regions corresponding to the dentin tissue. Therefore, in areas that correspond to the enamel tissue (side walls and occlusal), we used a material that is not fiber-reinforced and can be used in areas that are open to the oral environment.

In the MT and HGF groups, we did not use these fiber-reinforced composites, so we did not create side walls as a separate step in these groups. When restoring these groups, an automatrix band was placed on the tooth and the restoration was carried out layer by layer as in the control and resin composite groups.

Comments 6: Line 258 – add the type of alcohol

Response 6: The type of alcohol has been specified in the requested sentences.

Comments 7: Was the occlusal surface characteristics the same for all groups? If yes, how it was standardized? It could have had a significant influence on the results if in some samples the enamel was covered with composite resin and in others was not.

Response 7: The occlusal surface characteristics could not be prepared the same in the groups. However, as mentioned in the material and methods section, teeth with similar dimensions and morphologies (cusps, etc.) were used. In addition, the teeth were not beveled the enamel-restoration border was inspected under magnification, and care was taken not to cover the enamel with composite resin in any group. For the fracture test, a 30° inclined load was applied to the junction of the restoration and enamel on the buccal cusps.

Comments 8: Study group with typical reconstruction using post placed in root canal should be added.

Response 8: We also considered adding this group when planning experimental groups. However, considering the clinical scenario in a tooth with perforation in the furcal region, we thought placing a prefabricated post in the canal would not be appropriate. While preparing the additional post space, we were concerned that the cavity prepared with the drill would merge with the perforation area. While planning this study, we wanted to include techniques that could be applied to teeth with damaged PCD regions without additional preparation in this area. For this reason, in the Bioblock technique, we removed the root canal filling as if we were applying the retreatment technique. We did not prepare post space with a drill.

Comments 9: Study groups differed with regard to technique and materials used therefore it is impossible to indicate the reason for the difference in the results.

Response 9: In our preliminary study, before deciding on the experimental groups, we also wanted to use the Bioblock technique with the Everx posterior material.  However, since its consistency is denser, we had difficulty condensing it into the narrow mesial canals of the teeth. For this reason, we did not add it to our experimental group. We chose the modified transfixed and horizontal glass fiber post techniques because they are compatible with the no-post concept.

Comments 10: The cavity design is far different from the one performed in clinical scenarios.

Response 10: We tried to make the cavities as standard as possible and suitable for the clinical scenario.

Comments 10: Present figure 6 as box and whiskers plot.

Response 10: Figure 6 was added as box and whiskers plot to the manuscript.

Comments 11: Line 401-402: ‘Additionally, the authors reported that EXF showed more shrinkage-induced cuspal deformation than EXP.’ – add a reference or information how it was measured

Response 11: The requested reference was added to the manuscript.

Comments 12: Are fracture resistance values obtained for control and study groups observed in clinical conditions? What is clinical significance of the study?

Response 12: A paragraph on the clinical significance of the study was added to the discussion section.

Comments 13: Limitations of the study: what other examination methods could be employed for the further analysis? What are future perspectives?

Response 13: Sentences mentioning future analyses and perspectives were added to the discussion section. The first conclusion has been revised.

Comments 14: English grammar should be double checked, i.e. line 60, 74, etc.

Response 13: As per your request, the English grammar of the article has been checked. English editing has been made, and a document indicating that it has been done has been uploaded to the supplementary files.

Reviewer 2 Report

Comments and Suggestions for Authors

Dear authors:

In my opinion, this is an excellent manuscript in which the authors you present a study on the evaluation of short fiber-reinforced composite materials and fiber-reinforced restorations for endodontically treated molars with furcal perforation.

The manuscript is suitable for publication in the journal Polymers, as it will undoubtedly capture the interest of readers who seek to link the structure and properties of certain composite materials with improvements in dental treatment outcomes. The study aligns with the growing interest in minimally invasive endodontics, a field in which several studies have been conducted to explore the effects of access cavity design, root canal preparation protocols, and post-coring devices on endodontically treated teeth.

One of the manuscript’s key strengths is its groundbreaking approach to issues such as the effect of restoration techniques on fracture resistance in teeth with furcal perforation repaired with calcium silicate cement—an area that has not been tested before. Furthermore, as the study involves experiments with human teeth, you clearly specify that approval was obtained from the ethical committee of your university.

However, I suggest the following points to improve the manuscript:

  1. Clarify the source of the microfiber data mentioned in lines 74 and 75.
  2. The relative humidity value in line 185 is not specified.
  3. In line 206, the "2" in cm² should be written as a superscript.
  4. The first "Ever" in line 222 should begin with a capital letter.
  5. The values expressed in lines 306 to 308 and in the subsequent figure 6 lack clarity and should be rounded for better precision. For example, the first value could be written as 596 ± 45.
  6. It would be beneficial to include more specific terms related to the dental field in the keywords to improve the article's visibility and relevance to a broader audience.

In conclusion, this manuscript makes a valuable contribution to the field of endodontics and dental material science. Addressing the suggested revisions will further enhance its clarity and impact.

Author Response

Dear Reviewer,

Thank you for your valuable contribution. We believe that your comments and correction requests will improve both this study and our future work.

Comments 1: Clarify the source of the microfiber data mentioned in lines 74 and 75.

Response 1: The requested revisions and corrections have been made and highlighted in yellow.

Comments 2: The relative humidity value in line 185 is not specified.

Response 2: The requested revisions and corrections have been made and highlighted in yellow.

Comments 3: In line 206, the "2" in cm² should be written as a superscript.

Response 3: The requested revisions and corrections have been made and highlighted in yellow.

Comments 4: The values expressed in lines 306 to 308 and in the subsequent figure 6 lack clarity and should be rounded for better precision. For example, the first value could be written as 596 ± 45.

Response 4: The requested revisions and corrections have been made and highlighted in yellow.

Comments 5: It would be beneficial to include more specific terms related to the dental field in the keywords to improve the article's visibility and relevance to a broader audience.

Response 5: The requested revisions and corrections have been made and highlighted in yellow.

Round 2

Reviewer 1 Report

Comments and Suggestions for Authors

The manuscript has been significantly improved. Hence, some issues were not solved.

Materials and methods

Please add description of interproximal wall reconstruction in each study group as they may vary. The exact technique should be given.

Discussion

Line 410-411 ‘Additionally, the authors reported that EXF showed more shrinkage-induced cuspal deformation than EXP.’ – add a reference or information how it was measured. It has not been corrected in the previous review.

Author Response

Thank you for your valuable contributions. We have made the corrections you requested and highlight them in yellow.

Comments 1: Materials and methods- Please add description of interproximal wall reconstruction in each study group as they may vary. The exact technique should be given.

Response 1: The technique of interproximal wall build-up has been described in each study group. The sentences are highlighted in yellow.

Comments 2: Discussion- Line 410-411 ‘Additionally, the authors reported that EXF showed more shrinkage-induced cuspal deformation than EXP.’ – add a reference or information how it was measured. It has not been corrected in the previous review.

Response 2: A reference has been added to the requested sentence and highlighted in yellow.